# Sleep Disorders Associated with Neurodegenerative Diseases

**DOI:** 10.3390/diagnostics13182898

**Published:** 2023-09-10

**Authors:** Lucreția Anghel, Anamaria Ciubară, Aurel Nechita, Luiza Nechita, Corina Manole, Liliana Baroiu, Alexandru Bogdan Ciubară, Carmina Liana Mușat

**Affiliations:** 1Clinical Medical Department, Faculty of Medicine and Pharmacy, ‘Dunarea de Jos’ University, 800008 Galati, Romania; anghel_lucretia@yahoo.com (L.A.); anamburlea@yahoo.com (A.C.); nechitaaurel@yahoo.com (A.N.); nechitaluiza2012@yahoo.com (L.N.); lilibaroiu@yahoo.com (L.B.); 2‘Sf. Apostol Andrei’ Clinical Emergency County Hospital, 800578 Galati, Romania; carmina.musat@ugal.ro; 3‘Sf. Ioan’ Clinical Hospital for Children, 800487 Galati, Romania; 4‘Sf. Cuv. Parascheva’ Clinical Hospital of Infectious Diseases, 800179 Galati, Romania; 5Department of Morphological and Functional Sciences, Faculty of Medicine and Pharmacy, Dunarea de Jos’ University, 800008 Galati, Romania; bogdan.ciubara@ugal.ro

**Keywords:** sleep disturbances, neurological pathologies, amyotrophic lateral sclerosis (ALS), multiple system atrophy (MSA), hereditary ataxias, Huntington’s disease (HD), sleep-related breathing disorders, insomnia

## Abstract

Sleep disturbances are common in various neurological pathologies, including amyotrophic lateral sclerosis (ALS), multiple system atrophy (MSA), hereditary ataxias, Huntington’s disease (HD), progressive supranuclear palsy (PSP), and dementia with Lewy bodies (DLB). This article reviews the prevalence and characteristics of sleep disorders in these conditions, highlighting their impact on patients’ quality of life and disease progression. Sleep-related breathing disorders, insomnia, restless legs syndrome (RLS), periodic limb movement syndrome (PLMS), and rapid eye movement sleep behavior disorder (RBD) are among the common sleep disturbances reported. Both pharmacological and non-pharmacological interventions play crucial roles in managing sleep disturbances and enhancing overall patient care.

## 1. Introduction

Neurodegenerative diseases, a group of chronic and progressive disorders affecting the nervous system, pose a significant global health challenge. These conditions, including Alzheimer’s disease, dementia with Lewy bodies disease, Huntington’s disease, and others, are characterized by the gradual deterioration of nerve cells, leading to a decline in cognitive, motor, and functional abilities [1]. While the primary focus of research has been on the hallmark clinical features of these diseases, emerging evidence suggests a profound link between neurodegenerative conditions and sleep disturbances [2].

Sleep is a fundamental physiological process that plays a critical role in maintaining overall health and well-being. It serves as a time for restorative processes, memory consolidation, and the regulation of various bodily functions. However, individuals with neurodegenerative diseases frequently experience disrupted sleep patterns, such as insomnia [3], excessive daytime sleepiness, and abnormal sleep architecture. These sleep disturbances not only impact the quality of life for patients and caregivers but may also contribute to the progression and severity of the underlying neurodegenerative conditions.

The bidirectional relationship between sleep and neurodegeneration is a topic of growing interest and importance in both clinical and research settings. On one hand, the neurodegenerative processes themselves can directly affect the brain regions responsible for regulating sleep–wake cycles and sleep-related processes. On the other hand, sleep disturbances may accelerate or exacerbate neurodegeneration through mechanisms involving oxidative stress, inflammation, and impaired protein clearance [4]. Understanding the complex interplay between sleep and neurodegenerative diseases is crucial for uncovering potential therapeutic avenues and improving the overall management of these conditions.

This review aims to provide a comprehensive overview of the intricate relationship between sleep disorders and neurodegenerative diseases. By examining the current literature, exploring underlying mechanisms, and discussing potential implications, we seek to shed light on how sleep disturbances contribute to the pathophysiology of neurodegeneration and vice versa. Furthermore, we will explore the relevance of sleep-related interventions as a novel approach to managing and potentially slowing the progression of neurodegenerative diseases. In doing so, this review underscores the significance of recognizing sleep as a crucial factor in the holistic understanding and management of neurodegenerative disorders.

## 2. Sleep and Brain Anatomical Structures

The regulation of sleep and its underlying mechanisms are regulated by specific regions within the brain. In terms of microanatomy, it is commonly observed that the cell bodies of neurons responsible for synthesizing neurotransmitters involved in sleep mechanisms are typically concentrated in a specific region, while the terminal ends of their axons extend to other areas [5]. The cell bodies of the neurons implicated in sleep within the mammalian brain are situated in the brainstem, while their axons terminate in cerebral hemisphere centers. The process of sleep involves a structured interplay among the cerebral cortex, thalamus, and subcortical regions such as the brainstem. As stated in a study of Peplow et al. [6], the regulation of sleep and wakefulness in various regions of the brain is facilitated by the fluctuation of neurotransmitters in a controlled manner.

The hypothalamus is situated within the cerebral hemisphere, in close proximity to the pituitary gland. The hypothalamus is comprised of numerous nerve cell bodies known as the suprachiasmatic nuclei (SCN), which receive sensory input regarding light exposure in order to regulate the sleep and arousal cycle [7]. The pineal gland is situated within the anatomical recess located between the superior colliculi. The regulation of melatonin production, a neurohormone that promotes sleep, is influenced by various connections, thereby serving a crucial role in the regulation of the circadian rhythm. The amygdala, a neuroanatomical region implicated in the regulation of emotions, has been proposed to exhibit heightened activity during the phase of sleep characterized by rapid eye movement (REM sleep). This observation offers a potential explanation for the frequent comorbidity of mood disorders and disturbances in sleep patterns [8].

The neurons comprising the reticular activating system play a pivotal role in the regulation of wakefulness. The brain stem components, namely the midbrain, pons, and medulla oblongata, establish neural connections with the hypothalamus in order to regulate the circadian rhythm governing wakefulness and sleep. The midbrain is implicated in various physiological functions, including visual and auditory processing, motor coordination, regulation of sleep–wake cycles, maintenance of alertness, and control of body temperature. The pons and medulla oblongata exhibit specific associations with descending neural pathways responsible for the regulation of muscle activity, body posture, and limb movements during periods of relaxation. The brainstem nuclei implicated in the processing of sleep encompass the cholinergic nuclei situated at the junction of the pons and midbrain, the raphe nuclei, tuberomammillary nuclei, and locus coeruleus [5]. The thalamus serves as an intermediary hub for transmitting sensory information from the primary sense organs to the cerebral cortex. The ascending brainstem reticular activating system (ARAS) exhibits heightened activity during REM sleep, effectively transmitting signals at various intermediary locations, such as the thalamus. This activation serves to stimulate the forebrain during both wakefulness and REM sleep [9]. The thalamus is susceptible to impairments that can disrupt optimal brain functioning in humans, impacting both wakefulness and sleep [10]. The promotion of sleep has been attributed to the release of adenosine, a cellular energy byproduct, by cholinergic neurons located in the basal forebrain region. Caffeine and certain medications have been observed to mitigate drowsiness by inhibiting the effects of adenosine [11] (Figure 1).

## 3. Neurophysiology of Sleep

The concept of sleep is a multifaceted process that is intricately connected to neurological functioning. The central sleep and circadian regulation centers are situated in the intracranial region and encompass the anterior hypothalamus, reticular activating system, suprachiasmatic nucleus (SCN), and pineal gland. The regulation of sleep is commonly acknowledged to be influenced by the interplay between circadian and homeostatic mechanisms. The homeostatic mechanism of sleep pertains to the concept of “sleep drive,” which denotes the phenomenon where the inclination to sleep intensifies as the duration since the previous sleep period increases and diminishes as more time is spent accumulating sleep [12]. Sleep drive refers to the biological urge or pressure to sleep that accumulates over time as wakefulness is sustained. It is an essential component of our internal sleep regulation system and is primarily influenced by the length of time that has passed since the last period of sleep [13].

The longer we are awake, the stronger the sleep drive becomes. This drive to sleep gradually builds up as wakefulness continues, reflecting the body’s need for rest and recovery. It is part of the body’s way of maintaining a balance between wakefulness and sleep, ensuring that we obtain the rest we need to function optimally.

Sleep drive is regulated by several factors, including the body’s internal circadian rhythm (the natural body clock that regulates sleep–wake cycles); the amount of adenosine, a neurotransmitter that builds up during wakefulness and promotes sleep; and other complex biological mechanisms [14]. When sleep drive is high, it becomes increasingly difficult to stay awake, and eventually, the need for sleep becomes overwhelming.

The circadian timing system is responsible for the temporal organization of various neurobehavioral and physiologic processes, such as body temperature regulation, melatonin synthesis, and the 24 h sleep–wake cycle [15]. The suprachiasmatic nucleus (SCN) is a cluster of neurons situated in the inferior region of the hypothalamus, positioned slightly superior to the optic chiasm, which serves as the intersection point for the optic nerves. The suprachiasmatic nucleus (SCN) exhibits a high degree of sensitivity to light stimuli. The transmission of light through the retina initiates a pathway along the optic nerves towards the suprachiasmatic nucleus (SCN), subsequently stimulating the cessation of melatonin production by the pineal gland. Melatonin plays a crucial role in various physiological processes such as sleep regulation, thermoregulation, and blood pressure control. Its synthesis is most pronounced during the night-time period, characterized by reduced or absent exposure to light stimuli. The reticular activating system, situated in the midbrain, is primarily involved in sustaining a state of vigilance and attentiveness towards one’s surroundings, rather than directly regulating the sleep–wake cycle. Any disruptions occurring along this pathway have the potential to cause disturbances in the circadian rhythm and, consequently, sleep disturbances [16].

The process of sleep involves the activation and deactivation of specific brain structures during different stages of sleep [17]. Typical sleep is comprised of four stages of non-rapid eye movement (non-REM) sleep and one stage of rapid eye movement (REM) sleep, each exhibiting unique electroencephalogram (EEG) characteristics. The brain undergoes these phases in a cyclical manner, occurring at intervals of approximately 90 min, and typically repeating four to five times during a single night. Stage 1 represents the initial phase of the sleep cycle, characterized by the transition from wakefulness to sleep. During this stage, individuals are in a state of light sleep and can be easily roused. They may not have conscious awareness of having been asleep. During Stage 2 sleep, individuals experience a diminished state of conscious awareness accompanied by the emergence of distinct electroencephalogram patterns known as “sleep spindles” and “K complexes.” Additionally, there is a reduction in heart and respiratory rates, as well as a decline in body temperature. During Stages 3 and 4 of the sleep cycle, commonly referred to as deep sleep, there is a deceleration in brain wave activity, resulting in a diminished capacity for arousal. In the event that arousal is experienced, the individual may exhibit symptoms of grogginess and disorientation. Rapid eye movement (REM) sleep represents the concluding phase of the sleep cycle, characterized by the occurrence of dreams. During the rapid eye movement (REM) sleep stage, individuals experience rapid eye movements, elevated heart and respiratory rates, and frequently encounter muscle twitches. A decline in Stages 3 and 4 and REM sleep is observed as a typical manifestation of the aging process. This reduction in sleep stages may contribute to the prevalence of nocturnal awakenings, challenges in resuming sleep, and feelings of fatigue during the daytime that are frequently reported among older individuals.

## 4. The Link between Sleep and Neurological Disorders

The relationship between sleep and mental disorders is complex and multifaceted. Sleep disturbances, such as insomnia, hypersomnia, nightmares, and circadian rhythm disorders, frequently co-occur with various mental health conditions. Diagnostic criteria, as outlined in the Diagnostic and Statistical Manual of Mental Disorders (DSM-5), aid clinicians in identifying sleep disorders within the broader context of mental health evaluations. However, it is vital to recognize the bidirectional nature of this relationship, as untreated sleep disorders can exacerbate mental health symptoms and vice versa.

### 4.1. Sleep Disorders in Alzheimer’s Disease (AD)

Alzheimer’s disease (AD) is widely recognized as the prevailing neurodegenerative disorder, distinguished by the destruction of neurons and the accumulation of b-amyloid in neurofibrillary tangles within the hippocampus and cortex. These pathological changes ultimately result in cognitive impairment [18].

Both insufficient sleep duration and obstructive sleep apnea (OSA) have been identified as risk factors for Alzheimer’s disease (AD). The APOE-4 allele has been identified as a significant risk factor for both Alzheimer’s disease (AD) and obstructive sleep apnea (OSA) [19].

The relationship between Alzheimer’s disease (AD) and sleep difficulties is reciprocal. The condition leads to significant sleep deprivation and disruptions due to the degeneration of the suprachiasmatic and cholinergic nuclei. These abnormalities may potentially impact cognition by potentially impairing the brain’s ability to eliminate β-amyloid.

Sleep disturbances are prevalent, affecting around 45% of individuals, and have a substantial influence on both patients and their caretakers [20]. Sleep problems might potentially present as an initial indication; however, their occurrence and severity often escalate in tandem with the gravity of the underlying condition.

One prevalent sleep issue is an amplified inclination to advance sleep phases, which is distinguished by frequent daytime somnolence, challenges in initiating sleep at night, fragmented sleep throughout the night, and rising early in the morning [21].

The observed pattern in question has similarities to that reported in older individuals, but with greater severity. It is plausible that this pattern is largely attributable to the degradation of the suprachiasmatic nucleus, which subsequently leads to disturbances in the rhythm of melatonin production.

In some instances, individuals may display sundowning syndrome, which is distinguished by the manifestation of restlessness, cognitive impairment, and aggressive behavior throughout the evening and night-time hours. Patients may exhibit confusional arousals, sleepwalking, and agitation during nocturnal periods [22].

The results of polysomnography indicate a reduction in the overall duration of sleep and the effectiveness of sleep, an increase in the time it takes to fall asleep and the duration of wakefulness after falling asleep, a decrease in the duration of both deep sleep and REM sleep, and an increase in the duration of light sleep. Sleep recording in advanced instances presents challenges owing to the lack of the alpha rhythm while awake and the absence of sleep spindles and K complexes. The occurrence of rapid eye movement (REM) sleep disturbances is infrequent, as shown by research findings [23].

The prevalence of restless legs syndrome (RLS) does not seem to be noteworthy, maybe due to the underestimation of its frequency. This underestimation may arise from the diagnostic need for patients to articulate the sensations experienced in their legs. Dementia-specific criteria for restless legs syndrome (RLS) have been established [24].

The incidence of obstructive sleep apnea (OSA) is substantial, impacting a significant proportion of persons, ranging from 40% to 70%. Moreover, it has been shown that OSA might exacerbate cognitive impairment in patients with Alzheimer’s disease.

The treatment of sleep problems is founded upon cognitive–behavioral methods, sleep hygiene practices, and the use of bright light therapy [25].

During the first phases, the use of acetylcholinesterase inhibitors has shown the potential to ameliorate sleep disturbances and enhance cognitive function. Continuous positive airway pressure (CPAP) therapy is recommended for the treatment of obstructive sleep apnea (OSA) [26].

Despite the limited availability of compelling scientific data, many drugs are often used to enhance sleep quality. These include melatonin, benzodiazepines, sedative antidepressants, and atypical antipsychotics, such as quetiapine [27].

### 4.2. Sleep Disorders in Dementia with Lewy Bodies (DLB)

Dementia with Lewy bodies (DLB) is the second most prevalent kind of dementia, behind Alzheimer’s disease. It is distinguished by a combination of symptoms, including cognitive decline, parkinsonism, motor irregularities, and visual hallucinations. The neuropathological alterations include the manifestation of Lewy bodies throughout the brainstem, limbic system, and cortical regions.

Sleep disturbances are prevalent in almost 80% of individuals diagnosed with dementia with Lewy bodies (DLB), surpassing the occurrence rate seen in Alzheimer’s disease (AD) [28].

Insomnia, a circadian rhythm disorder characterized by early morning waking, is often accompanied with excessive daytime sleepiness (EDS) resulting from frequent napping. Additionally, individuals with this condition may have nocturnal hallucinations and confusional arousals accompanied by sleepwalking, which are often-seen symptoms.

Episodes of confusional arousals often manifest during the non-rapid eye movement (non-REM) sleep phases.

There is no evidence to suggest that DLB patients have a higher prevalence of obstructive sleep apnea (OSA), periodic limb movement disorder (PLMD), and restless legs syndrome (RLS) compared to those of a comparable age in the general community [24].

Individuals diagnosed with idiopathic rapid eye movement sleep behavior disorder (RBD) may exhibit a range of cognitive impairments, including modest deficits in executive functioning, visuospatial abilities, and memory.

The temporal duration between moderate cognitive impairment and the initiation of dementia is estimated to be roughly 2 years.

Rapid eye movement behavior disorder (RBD) has been seen in around 80% of patients and has been shown to potentially precede the start of dementia in around 70% of cases [29].

The identification of rapid eye movement behavior disorder (RBD) in an individual with dementia is indicative of dementia with Lewy bodies (DLB), since this sleep disorder rarely occurs in alternative types of dementia such as Alzheimer’s disease (AD), frontotemporal dementia, and progressive supranuclear palsy.

The current diagnostic criteria for dementia with Lewy bodies (DLB) include idiopathic rapid eye movement sleep behavior disorder (RBD) as an indicative characteristic of the condition.

Idiopathic rapid eye movement sleep behavior disorder (RBD) is mostly seen in males within the context of dementia with Lewy bodies (DLB). This condition is characterized by a brief duration of dementia, an early start of parkinsonism, visual hallucinations, and the presence of non-Alzheimer’s pathology upon postmortem examination.

Differentiating between confusional arousals and nocturnal visual hallucinations from REM sleep behavior disorder (RBD) merely based on the patient’s clinical history might pose challenges.

Individuals diagnosed with advanced dementia who undergo polysomnography may have disturbances in their sleep patterns as a result of the deterioration of neurological systems responsible for regulating sleep [30].

### 4.3. Sleep Disorders in Progressive Supranuclear Palsy (PSP)

Progressive supranuclear palsy (PSP) is a neurodegenerative disorder that is distinguished by a combination of clinical symptoms, including dementia, parkinsonism, falls, and vertical gaze palsy.

Progressive supranuclear palsy (PSP) is a neurological condition characterized by the degeneration of many regions within the brain, including the brainstem, basal ganglia, frontal lobe, and other associated areas [31].

Polysomnographic investigations have shown a decline in overall sleep duration, a fall in the proportion of rapid eye movement (REM) sleep, as well as a reduction in the occurrence of sleep spindles and K complexes.

In instances of significant severity, the alpha rhythm is not present, resulting in challenges in distinguishing between wakefulness and sleep phases.

Excessive daytime sleepiness (EDS), obstructive sleep apnea syndrome (OSAS), periodic limb movement disorder (PLMD), and restless legs syndrome (RLS) are seldom-seen consequences in progressive supranuclear palsy (PSP).

Insomnia is the prevailing sleep disturbance seen in individuals with progressive supranuclear palsy (PSP). It is characterized by challenges in both beginning and sustaining sleep, sometimes accompanied with indications indicative of rapid eye movement sleep behavior disorder (RBD) [29].

In some individuals, symptoms resembling rapid eye movement sleep behavior disorder (RBD) may manifest as episodes of nocturnal roaming and confusional arousals. These manifestations are often seen in patients diagnosed with various forms of dementia.

Rapid eye movement sleep behavior disorder (RBD) manifests in roughly 10% to 20% of individuals diagnosed with progressive supranuclear palsy (PSP) and is often characterized by a modest intensity. This condition frequently emerges subsequent to the beginning of dementia.

Approximately 20% of patients exhibit the subclinical manifestation of REM sleep behavior disorder (RBD), characterized by the absence of symptoms and the persistence of REM sleep without atonia.

### 4.4. Sleep Disorders in Huntington’s Disease (HD)

Huntington’s disease (HD) is a hereditary condition that follows an autosomal dominant pattern of inheritance. It is distinguished by symptoms such as dementia, chorea, and mental disorders. These symptoms are linked to the presence of enlarged CAG repeats in the Huntington gene [32].

Pathological investigations reveal significant degeneration of the striatum, including the caudate nucleus and putamen, with a comparatively lower impact on the cortex [33].

Sleep difficulties have been shown to impact a significant proportion, specifically up to 87%, of those diagnosed with Huntington’s disease (HD), with a greater prevalence seen in those in late stages of the illness. The incidence of sleep disturbances rises in proportion to the intensity and duration of the ailment.

It is often seen among patients that they have suboptimal sleep quality, insomnia, fragmented sleep patterns with many awakenings throughout the night, excessive daytime sleepiness (EDS), and a circadian rhythm disruption defined by an advanced sleep phase leading to early morning waking [34].

Polysomnographic investigations have shown a decrease in sleep efficiency, an increase in wakefulness after sleep initiation, an elevated proportion of light sleep, a lengthening of the time it takes to enter rapid eye movement (REM) sleep, and a reduction in the proportion of deep sleep and REM sleep.

Rapid eye movement sleep behavior disorder (RBD) [29], obstructive sleep apnea syndrome (OSAS) [26], and restless legs syndrome (RLS) [24] have been seen in individuals with Huntington’s disease (HD); however, their occurrence is neither prevalent nor deemed of any concern.

Individuals with the premutation gene have a higher degree of sleep disruption, defined by a fragmented sleep pattern, as compared to control participants.

### 4.5. Sleep Disorders in Hereditary Ataxias

Hereditary ataxias are a class of neurological illnesses that are often transmitted with genetic inheritance, mostly caused by mutations in certain genes [35].

These illnesses have an impact on many brain areas such as the spinocerebellar tracts, cerebellum, and brainstem. Clinically, these individuals have increasing ataxia along with a diverse range of other neurological symptoms and indications, including polyneuropathy and parkinsonism [36].

Hereditary ataxias are not often associated with obstructive sleep apnea syndrome (OSAS) and excessive daytime sleepiness (EDS).

The occurrence of nocturnal stridor resulting from vocal cord anomalies has been documented in spinocerebellar ataxias (SCA) 1 and SCA 3 [37].

Rapid eye movement sleep behavior disorder (RBD) has been seen in a significant proportion, around 50%, of individuals diagnosed with spinocerebellar ataxia type 3 (also known as Machado–Joseph disease). Additionally, patients with spinocerebellar ataxia type 2, although with moderate clinical severity, have also been shown to exhibit RBD. The presence of RBD has not been detected in individuals with spinocerebellar ataxia type 2 (SCA 2).

Sleep-related breathing difficulties have been documented in spinocerebellar ataxia type 1 (SCA 1), spinocerebellar ataxia type 2 (SCA 2), spinocerebellar ataxia type 3 (SCA 3), and spinocerebellar ataxia type 6 (SCA 6).

### 4.6. Sleep Disorders in Amyotrophic Lateral Sclerosis (ALS)

Amyotrophic lateral sclerosis (ALS) is distinguished by the degeneration of neurons situated in the cortex, brainstem nuclei, and the ventral horn of the spinal cord [38].

The condition exhibits a constellation of symptoms including rapidly advancing weakness, muscular wasting, involuntary muscle twitches, increased muscle tone, difficulty with speech, difficulty with swallowing, and the inability to breathe properly.

Respiratory dysfunction arises as a consequence of the weakening of the pharyngeal, laryngeal, diaphragmatic, and intercostal muscles due to the depletion of motor neurons in the brainstem and spinal cord. Apnea is the most prevalent sleep problem.

Sleep-related respiratory disorders occur in 17% to 76% of cases and include nocturnal hypoventilation, obstructive sleep apnea syndrome (OSAS) [26], laryngeal stridor, and central sleep apnea. One prevalent manifestation is nocturnal hypoventilation, which arises from the paralysis of the diaphragmatic, intercostal, and accessory respiratory muscles [39].

The incidence of sleep apnea is lower compared to hypoventilation. The majority of apneic occurrences are classified as central in nature due to the presence of diaphragm weakness and paralysis of the accessory respiratory muscles, which contribute to the development of this particular kind of sleep apnea.

To manage respiratory compromise in ALS, various interventions are employed. CPAP (continuous positive airway pressure) is commonly used to alleviate obstructive sleep apnea and ensure adequate airflow during sleep [40]. Non-invasive ventilation (NIV) is another crucial approach, providing positive pressure support to maintain proper oxygen levels and assist breathing. NIV devices, such as bilevel positive airway pressure (BiPAP) machines, are designed to support the weakened respiratory muscles of ALS patients [41].

In some regions, particularly in Europe, Italy, and East Asian countries, more advanced interventions may be considered. Tracheostomy or tracheostomy-assisted ventilation might be employed for those who require more sustained and invasive respiratory support. Tracheostomy allows for direct access to the airway, enabling long-term mechanical ventilation through a tracheostomy tube [42].

The use of these interventions can significantly impact the sleep quality and overall well-being of ALS patients. While respiratory support measures aim to enhance breathing, they can also influence sleep architecture and potentially cause sleep disturbances. Ensuring patient comfort, optimizing ventilator settings, and offering psychological support are vital aspects of managing sleep-related issues in ALS patients undergoing such interventions.

The primary sleep problems are nocturia, muscular cramps, early insomnia, decreased total sleep duration, and excessive daytime sleepiness (EDS) [43].

Rapid eye movement sleep behavior disorder (RBD) [44] and restless legs syndrome (RLS) [43] are infrequent occurrences that may manifest in certain instances.

### 4.7. Sleep Disorders in Multiple System Atrophy (MSA)

Multiple system atrophy (MSA) is a neurological disorder that manifests as a mix of parkinsonism, cerebellar syndrome, and autonomic failure, occurring in diverse combinations. The defining pathological feature of MSA is the presence of alpha-synuclein-positive cytoplasmic inclusions in the glial cells of different brain areas [45]. Sleep issues are reported by around 70% of people diagnosed with multiple system atrophy (MSA).

Approximately 50% of individuals have insomnia during sleep onset and interrupted sleep. There are many potential variables that might lead to sleep fragmentation. These include urine incontinence, anxiety, sadness, the difficulty to change body position in bed owing to parkinsonism, and the use of several drugs [46].

Excessive daytime sleepiness (EDS) is seen in around 28% of individuals, but it is often not considered a prevalent condition. Sleep attacks may occur in some people after the treatment of levodopa [47].

The prevalence of restless legs syndrome (RLS) and periodic limb movement syndrome (PLMS) in the general population of a comparable age remains uncertain.

Multiple system atrophy (MSA) is found to be ultimately identified in a limited number of individuals who were initially diagnosed with idiopathic rapid eye movement sleep behavior disorder (RBD). It is worth noting that a majority of these individuals are subsequently diagnosed with Parkinson’s disease and dementia with Lewy bodies. This discrepancy in diagnosis rates can be attributed to the relatively lower prevalence of MSA in the general population when compared to Parkinson’s disease and dementia with Lewy bodies.

The occurrence of rapid eye movement sleep behavior disorder (RBD) in patients with multiple system atrophy (MSA) ranges from 80% to 100% [29]. The lack of rapid eye movement sleep behavior disorder (RBD) in a patient suspected of having multiple system atrophy (MSA) raises significant doubts about the accuracy of the diagnosis for this particular condition. The inclusion of RBD in the diagnostic criteria for multiple system atrophy (MSA) is now recognized. Rapid eye movement sleep behavior disorder (RBD) has been seen to occur prior to the manifestation of the cardinal symptoms of multiple system atrophy (MSA) in around 50% of individuals [48].

The manifestation of respiratory difficulties in individuals with multiple system atrophy (MSA) may arise from both central and peripheral sources, including obstructive factors. Degeneration of the bulbar respiratory centers is the primary cause of central respiratory disorders. This degeneration results in many aberrant respiratory responses, including hypoxic ventilatory responses, Cheyne–Stokes breathing, irregular breathing, and central sleep apnea [39].

Nocturnal stridor, a high-pitched sound produced with disrupted airflow during sleep, is a common sleep-related issue in MSA patients. Its occurrence can vary depending on the stage of MSA and individual patient factors. It is noteworthy that nocturnal stridor appears to be relatively frequent in MSA patients, and the reported prevalence may vary between different studies and clinical practices [49].

Approximately 20% of patients have nocturnal stridor.The prevalence of the phenomenon discussed in the previous statement may vary depending on the stage of MSA and geographical differences, perhaps exceeding the original estimate provided [50]. Based on clinical findings, it has been noted that a significant proportion of individuals with multiple system atrophy (MSA) encounter this particular symptom at some stage along the progression of their illness. The presence of this particular indicator is regarded as a noteworthy signal that ought to elicit skepticism about the potential presence of multiple system atrophy (MSA) in a patient exhibiting parkinsonism.

Stridor is seen throughout all phases of the ailment and signifies the presence of an airway blockage specifically at the level of the vocal cords inside the larynx. As the condition advances, there is a possibility of experiencing nocturnal stridor during waking as a result of the progressive constriction of the glottis [51].

Individuals diagnosed with multiple system atrophy (MSA), particularly those exhibiting stridor, may encounter characteristic occurrences of obstructive sleep apnea (OSA) accompanied by oxygen desaturations. Additionally, in some instances, they may face subacute bouts of respiratory failure [52].

The occurrence of vocal cord paralysis is often found in a significant proportion of patients who exhibit nocturnal stridor. This observation is made with clinical evaluation of the vocal cords using laryngoscopy when the patients are awake.

The occurrence of stridor in multiple system atrophy (MSA) has been shown to be associated with reduced rates of survival and an elevated likelihood of experiencing unexpected death while asleep [51].

The use of continuous positive airway pressure (CPAP) and tracheostomy has been shown to effectively alleviate symptoms of stridor and obstructive apneas in individuals with multiple system atrophy (MSA) (Table 1).

## 5. Treatment

The management of sleep disturbances in neurodegenerative diseases is a complex endeavor, often requiring a tailored approach that considers the underlying disease pathology and the specific sleep-related symptoms experienced by patients. Below, we outline various treatment methods and interventions with their corresponding relevance to different neurodegenerative diseases and pathological conditions.

### 5.1. Pharmacological Interventions

Pharmacological approaches can target specific sleep disturbances commonly associated with neurodegenerative diseases. For instance, patients with Parkinson’s disease experiencing REM sleep behavior disorder (RBD) might benefit from medications that suppress REM sleep, such as Temazepam or melatonin. Individuals with Alzheimer’s disease facing insomnia could be prescribed sedative-hypnotics, although caution is exercised due to potential cognitive side effects [73].

Pharmacological approaches for managing sleep disturbances can vary based on the specific neurological pathology and type of insomnia observed:Benzodiazepines: Estazolam, Quazepam, Triazolam, Flurazepam, Temazepam;Non-benzodiazepines: Zaleplon, Zolpidem, Eszopiclone;Sedative Antidepressants: Doxepin;Melatonin Receptor Agonists: Ramelteon;Melatonin.

These pharmacological options can be relevant for patients across different neurodegenerative diseases based on the predominant sleep-related symptoms they experience.

### 5.2. Non-Pharmacological Interventions

Non-pharmacological interventions play a crucial role in managing sleep disturbances, often focusing on improving sleep hygiene and behavioral modifications. Patients across various neurodegenerative diseases can benefit from creating a consistent sleep schedule, optimizing the sleep environment, and engaging in relaxation techniques [74]. Cognitive–behavioral therapy for insomnia (CBT-I) has proven effective in addressing insomnia in neurodegenerative diseases by targeting maladaptive sleep-related behaviors and thoughts [75].

Non-pharmacological interventions also vary depending on the neurological pathology and type of insomnia:External Devices for Assisting Breathing (CPAP—Continuous Positive Airway Pressure): Particularly relevant for patients with sleep-related breathing disorders, such as obstructive sleep apnea, which can occur in various neurodegenerative diseases.Phototherapy: Valuable for conditions like Alzheimer’s disease, where disruptions in circadian rhythms are common and can contribute to sleep disturbances.Cognitive–Behavioral Therapy: Effective in addressing insomnia in neurodegenerative diseases by targeting maladaptive sleep-related behaviors and thoughts.Sleep Hygiene Practices: A general approach that can benefit patients across various neurodegenerative diseases by promoting healthy sleep habits.

### 5.3. Continuous Positive Airway Pressure (CPAP) and Non-Invasive Ventilation

Patients with neurodegenerative diseases that exhibit sleep-related breathing disorders, such as obstructive sleep apnea, might be candidates for CPAP or non-invasive ventilation. This intervention is particularly relevant for conditions like multiple system atrophy (MSA) where nocturnal stridor is frequent, and ALS where respiratory muscle weakness leads to compromised breathing during sleep [76].

### 5.4. Light Therapy

Light therapy has shown promise in regulating sleep–wake cycles, particularly in conditions like Alzheimer’s disease where disruptions in circadian rhythms are common. Exposure to bright light during specific times of the day can help re-establish a proper sleep–wake pattern and alleviate sleep disturbances [77].

### 5.5. Management of REM Sleep Behavior Disorder (RBD)

In cases of REM sleep behavior disorder, where patients physically act out their dreams during REM sleep, safety measures are vital. This may involve creating a safe sleep environment by removing potentially harmful objects from the bedroom [78].

### 5.6. Addressing Restless Legs Syndrome (RLS)

Patients with restless legs syndrome, common in neurodegenerative diseases like Parkinson’s disease, might benefit from iron supplementation and dopamine agonist medications. Treating the underlying condition contributing to RLS can also alleviate its symptoms [79].

It is important to note that treatment approaches should be individualized, considering the patient’s overall health, disease stage, and specific sleep-related symptoms. Multidisciplinary collaboration involving neurologists, sleep specialists, psychologists, and other healthcare professionals is often necessary to optimize treatment strategies for sleep disturbances in the context of neurodegenerative diseases.

## 6. Discussions

The present study aimed to investigate the sleep disorders associated with various neurological pathologies, including amyotrophic lateral sclerosis (ALS), multiple system atrophy (MSA), hereditary ataxias, Huntington’s disease (HD), progressive supranuclear palsy (PSP), and dementia with Lewy bodies (DLB) [80]. The findings shed light on the prevalence and characteristics of sleep disturbances in these conditions, contributing to a better understanding of their impact on patients’ quality of life and disease progression.

In ALS, sleep-related breathing disorders, particularly hypoventilation and central sleep apnea, were observed in a significant proportion of patients [43]. These respiratory disturbances can lead to serious complications and may contribute to the rapid disease progression. Identifying and managing these sleep-related issues are crucial in the overall care of ALS patients.

In MSA, sleep problems were reported by a substantial number of patients, with insomnia, fragmented sleep, and excessive daytime sleepiness being the most common issues [47]. Stridor, a prominent sign of MSA, emerged as a significant predictor of decreased survival and an increased risk of sudden death during sleep. Non-pharmacological interventions like continuous positive airway pressure (CPAP) and tracheostomy have shown efficacy in alleviating stridor and obstructive apneas, thus providing potential avenues for symptom management.

Hereditary ataxias are associated with various sleep disturbances, including insomnia, restless legs syndrome (RLS), and periodic limb movement syndrome (PLMS) [81]. The exact prevalence of RLS and PLMS in hereditary ataxias remains unclear, warranting further investigation. However, it is evident that these sleep disorders significantly impact patients’ sleep quality and may exacerbate existing motor and cognitive symptoms.

Huntington’s disease is characterized by various sleep issues, such as insomnia at sleep onset and fragmented sleep, which can be influenced by several contributing factors, including anxiety, depression, and medication use. Proper management of these sleep disturbances may improve patients’ overall well-being and potentially delay disease progression [82].

In progressive supranuclear palsy (PSP), sleep disorders, such as rapid eye movement sleep behavior disorder (RBD), are prevalent. RBD emerged as a valuable diagnostic criterion for PSP, as its absence can raise suspicion of other neurodegenerative conditions [44]. Early detection of RBD may enable earlier intervention and improved disease management.

Dementia with Lewy bodies (DLB) is associated with rapid eye movement sleep behavior disorder (RBD) in a significant number of patients. This finding reinforces the importance of considering RBD as a potential early marker of DLB and highlights its diagnostic significance in differentiating DLB from other forms of dementia [60].

Overall, the findings of this study underscore the importance of recognizing and addressing sleep disorders in various neurological pathologies. Both pharmacological and non-pharmacological interventions play vital roles in managing these sleep disturbances and enhancing patients’ overall quality of life. Further research is warranted to better understand the underlying mechanisms of sleep disturbances in neurological diseases, thus paving the way for more effective and tailored treatment strategies. Integrating sleep assessment and management into the comprehensive care of patients with neurological disorders is crucial to optimize their clinical outcomes and overall well-being.

## 7. Future Perspectives

### 7.1. Sleep and Digital Health Interventions

Virtual Reality (VR) and Sleep Therapy: Future developments in virtual reality technology may enable the creation of immersive and interactive sleep therapy experiences. Virtual environments could be tailored to address specific sleep disturbances related to different mental disorders, providing a unique and engaging approach to sleep interventions.

Digital Sleep Tracking and Feedback: Advancements in digital sleep tracking and artificial intelligence algorithms could lead to personalized sleep feedback systems. Integrating sleep data from wearable devices and other sources, AI-powered platforms may offer real-time insights and actionable recommendations for optimizing sleep in individuals with mental disorders.

### 7.2. Sleep as a Target for Novel Treatment Approaches

Targeting Sleep Circuitry: Emerging research on the neural circuits involved in sleep regulation and their interactions with mental health pathways may open new avenues for therapeutic interventions. Targeted neuromodulation techniques, such as transcranial magnetic stimulation (TMS) or deep brain stimulation (DBS), could be explored to modulate sleep-related brain regions in mental disorders.

Chronotherapeutics: Chronotherapy, the strategic timing of treatments based on circadian rhythms, could be investigated as a complementary approach for managing sleep disturbances in mental disorders. Tailoring interventions to align with individual circadian preferences may optimize treatment outcomes.

### 7.3. Sleep and Gut–Brain Axis

Gut Microbiome and Sleep: The gut–brain axis, a bidirectional communication system between the gut microbiome and the central nervous system, has emerged as a fascinating area of research in sleep and mental health. Studies suggest that gut microbiota may influence sleep patterns through various mechanisms, including the production of neurotransmitters and metabolites that impact sleep regulation. Alterations in the gut microbiome have been associated with sleep disturbances and mental health conditions. Thus, interventions targeting the gut microbiome, such as probiotics or dietary changes, could potentially improve sleep quality and mental health outcomes.

Sleep-Inducing Nutraceuticals: Investigating natural sleep-inducing compounds presents a promising avenue for addressing sleep disruptions in mental disorders. Certain foods or herbal supplements rich in melatonin, a hormone that regulates sleep–wake cycles, might have sleep-promoting effects. Incorporating sleep-inducing nutraceuticals into dietary interventions could offer an adjunctive approach to improve sleep quality. However, further research is needed to determine the safety and efficacy of these compounds and their potential interactions with medications used in mental health treatment.

### 7.4. Sleep and Artificial Intelligence

AI-Powered Sleep Disorder Prediction: Artificial intelligence algorithms have shown remarkable capabilities in analyzing vast amounts of data. In the context of sleep and mental health, AI-powered platforms may integrate data from wearable devices, electronic health records, genetic information, and behavioral patterns to predict the likelihood of developing mental disorders based on sleep disturbances. Early identification of individuals at risk could facilitate timely interventions, preventive strategies, and personalized treatment plans. AI-driven approaches hold the potential to enhance the accuracy and efficiency of clinical assessments and improve patient outcomes.

In summary, the future perspectives for the aspects of sleep in mental disorders are exciting and multi-faceted. Digital health interventions, advances in sleep circuitry targeting, gut–brain axis research, and the integration of artificial intelligence all offer promising opportunities to optimize sleep interventions, enhance mental health care, and improve the well-being of individuals living with mental disorders. As these innovative approaches continue to evolve, the potential for transformational advancements in sleep research and mental health care becomes increasingly tangible.

## Figures and Tables

**Figure 1 diagnostics-13-02898-f001:**
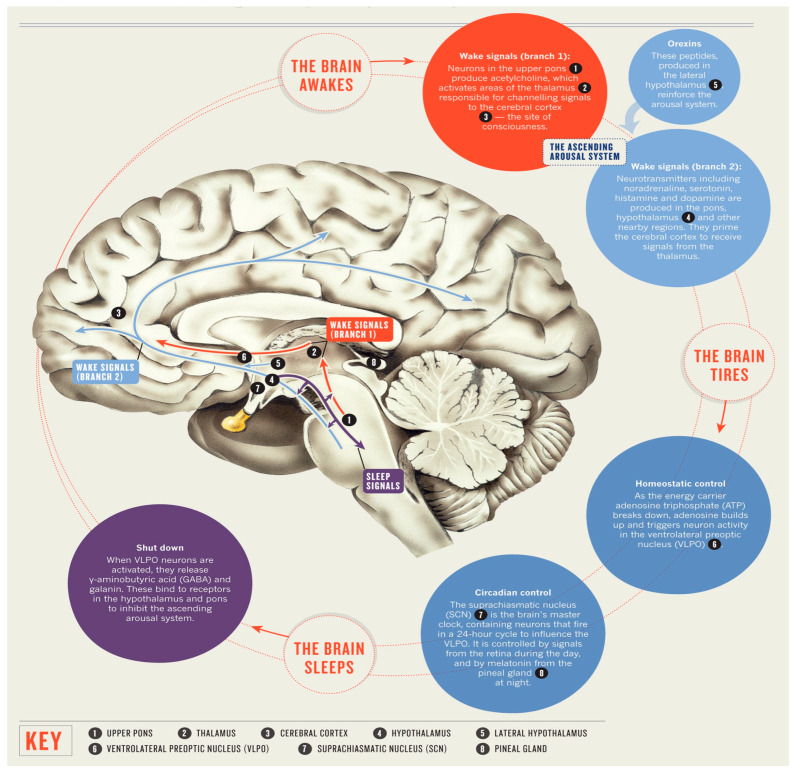
The anatomy of sleep.

**Table 1 diagnostics-13-02898-t001:** Studies Investigating Sleep Disturbances in Neurodegenerative Diseases.

Study	Study Design	Diseases Investigated	Patients (*n*)	Age Range	Gender Distribution (M/F)	Severity	Sleep Characteristics
Borges et al. [53]	Systematic review	Alzheimer	6912	40–91 years	45%/55%	Mild	Increased sleep fragmentation Longer sleep time latency
Mullins et al. [54]	Perspective	Alzheimer	100	60–85 years	64.7%/35.3%	Mild	Fragmented sleep Obstructive sleep apnea
Casagrande et al. [55]	Systematic review	Alzheimer	71	58–93 years	36%/64%	Severe	Repetitive awakenings Insomnia Higher sleep latency
Huo et al. [56]	Perspective	Alzheimer	96	62–75 years	49%/51%	Mild	Insomnia
Louzada et al. [57]	Clinical trial	Alzheimer	62	80 years	48%/52%	Mild	Insomnia
Sylwia et al. [58]	Study	Alzheimer	112	51 years	85%/15%	Severe	Sleep apnea
Kim et al. [59]	Study	Alzheimer	351	72–80 years	49.6%/50.4%	Mild	Insomnia
Chan et al. [60]	Systematic review	Dementia with Lewy bodies	83	>65 years	26%/74%	Mild	Repeated episodes of sleep
Koren et al. [61]	Systematic review	Dementia with Lewy bodies	90	66–91 years	N/A	Mild	Hyposmia REM sleep behavior disorder
Gaig et al. [62]	Retrospective	Progressive supranuclear palsy	22	46–83 years	50%/50%	Severe	Parasomnia, sleep apnea, insomnia, excessive daytime sleepiness
Chaithra et al. [63]	Prospective	Progressive supranuclear palsy	76	62–70 years	53%/47%	Severe	Excessive daytime sleepiness, obstructive sleep apnea syndrome
De Bruin et al. [64]	Prospective	Progressive supranuclear palsy	11	52–70 years	25%/75%	Severe	Insomnia
Sabater et al. [65]	Case series	Progressive supranuclear palsy	8	52–76 years	40%/50%	Severe	Obstructive sleep apnea
Pao et al. [66]	Retrospective	Progressive supranuclear palsy	78	71–80 years	85%/15%	Mild	Insomnia
Zhang et al. [67]	Systematic review	Huntington	152	43–57 years	N/A	Mild	Irregular sleep–wake cycles Decreased REM Increased REM latency
Maffi et al. [68]	Perspective	Huntington	42	28–64 years	52.4%/47.6%	Severe	Insomnia Altered sleep quality
Postuma et al. [69]	Clinical trial	Multiple system atrophy	1280	66–74 years	82.5%/17.5%	Severe	Fragmented sleep insomnia
De Pablo et al. [70]	Retrospective	Multiple system atrophy	111	62 years	60.4%/39.6%	Mild	Obstructive sleep apnea
Elliot et al. [71]	Clinical trial	Multiple system atrophy	361	>18 years	80%/20%	Mild	Insomnia
Wilke et al. [72]	Observational	Multiple system atrophy	23	22–67 years	45%/55%	Mild	Increased daytime sleepiness

## Data Availability

Data are contained within the article.

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
