# Peer review of "Sleep Disorders Associated with Neurodegenerative Diseases"

_diagnostics, 2023, doi:10.3390/diagnostics13182898_

Round 1
Reviewer 1 Report
Dear authors,
thank you for the opportunity to read and review the manuscript. The topic is interesting.
The paper is well written, it reports an interesting overview of anatomical and neurophysiological patterns of sleep; the figure is clear and interesting.
The title is not consistent with the text and with the aim of the review. Indeed, mental diseases and neurological diseases do not refer to the same pathologies.
In introduction section you refer to the link between sleep and mental disorders, but the aim of the paper is to investigate the link between neurological disorders (AD, DLB, PSP, HD, hereditary ataxias, ALS, MSA) and sleep, as you interestingly report in paragraph 4.
I think that throughout the paper there is a confusing use of mental disorder and neurological disorders as synonymous. Please improve.
in paragraph 2 the statement with reference six should be corrected with, e.g., “As stated by the study of Peplow et al.. […]” and not with citation number.
The figure is interesting.
In paragraph 3, “sleep drive” explanation is not clear and reference is missing.
Paragraph 4.1 reference is missing regarding the REM sleep disorders in AD.
Paragraph 4.6 reference is missing regarding RBD and RLS in ALS.
Table 1 is not clear, the title is not correct as refers to mental disorders and not neurological disorders, in the table disease are missing; the diseases investigated in the cited studies should be reported in the table to make it clearer and readable.
References are missing in the treatment section.
Author Response
First of all, we would like to thank you very much for your review and consideration of our work. Thank you!
Here are our answers to the subjects you highlighted:
The title is not consistent with the text and with the aim of the review. Indeed, mental diseases and neurological diseases do not refer to the same pathologies.
In introduction section you refer to the link between sleep and mental disorders, but the aim of the paper is to investigate the link between neurological disorders (AD, DLB, PSP, HD, hereditary ataxias, ALS, MSA) and sleep, as you interestingly report in paragraph 4.
I think that throughout the paper there is a confusing use of mental disorder and neurological disorders as synonymous. Please improve.
Response: We modified title and introduction section, highlighted in red.
Neurodegenerative diseases, a group of chronic and progressive disorders affecting the nervous system, pose a significant global health challenge. These conditions, including Alzheimer's disease, dementia with Lewy bodies disease, Huntington's disease, and others, are characterized by the gradual deterioration of nerve cells, leading to a decline in cognitive, motor, and functional abilities [1]. While the primary focus of research has been on the hallmark clinical features of these diseases, emerging evidence suggests a profound link between neurodegenerative conditions and sleep disturbances [2].
Sleep is a fundamental physiological process that plays a critical role in maintaining overall health and well-being. It serves as a time for restorative processes, memory consolidation, and the regulation of various bodily functions. However, individuals with neurodegenerative diseases frequently experience disrupted sleep patterns, such as insomnia [3], excessive daytime sleepiness, and abnormal sleep architecture. These sleep disturbances not only impact the quality of life for patients and caregivers but may also contribute to the progression and severity of the underlying neurodegenerative conditions.
The bidirectional relationship between sleep and neurodegeneration is a topic of growing interest and importance in both clinical and research settings. On one hand, the neurodegenerative processes themselves can directly affect the brain regions responsible for regulating sleep-wake cycles and sleep-related processes. On the other hand, sleep disturbances may accelerate or exacerbate neurodegeneration through mechanisms involving oxidative stress, inflammation, and impaired protein clearance [4]. Understanding the complex interplay between sleep and neurodegenerative diseases is crucial for uncovering potential therapeutic avenues and improving the overall management of these conditions.
This review aims to provide a comprehensive overview of the intricate relationship between sleep disorders and neurodegenerative diseases. By examining the current literature, exploring underlying mechanisms, and discussing potential implications, we seek to shed light on how sleep disturbances contribute to the pathophysiology of neurodegeneration and vice versa. Furthermore, we will explore the relevance of sleep-related interventions as a novel approach to managing and potentially slowing the progression of neurodegenerative diseases. In doing so, this review underscores the significance of recognizing sleep as a crucial factor in the holistic understanding and management of neurodegenerative disorders.
in paragraph 2 the statement with reference six should be corrected with, e.g., “As stated by the study of Peplow et al.. […]” and not with citation number.
Response: We corrected the sentence, highlighted in red.
As stated by the study of Peplow et al. [6], the regulation of sleep and wakefulness in various regions of the brain is facilitated by the fluctuation of neurotransmitters in a controlled manner.
In paragraph 3, “sleep drive” explanation is not clear and reference is missing.
Response: We added the following paragraphs and the missing reference [12], highlighted in red.
Sleep drive refers to the biological urge or pressure to sleep that accumulates over time as wakefulness is sustained. It's an essential component of our internal sleep regulation system and is primarily influenced by the length of time that has passed since the last period of sleep [13].
The longer we are awake, the stronger the sleep drive becomes. This drive to sleep gradually builds up as wakefulness continues, reflecting the body's need for rest and recovery. It's part of the body's way of maintaining a balance between wakefulness and sleep, ensuring that we get the rest we need to function optimally.
Sleep drive is regulated by several factors, including the body's internal circadian rhythm (the natural body clock that regulates sleep-wake cycles), the amount of adenosine, a neurotransmitter that builds up during wakefulness and promotes sleep, and other complex biological mechanisms [14]. When sleep drive is high, it becomes increasingly difficult to stay awake, and eventually, the need for sleep becomes overwhelming.
Paragraph 4.1 reference is missing regarding the REM sleep disorders in AD.
Response: We added the missing reference [23].
Paragraph 4.6 reference is missing regarding RBD and RLS in ALS.
Response: We added the missing references, [44], [43], highlighted in red.
Rapid eye movement sleep behavior disorder (RBD) [44] and restless legs syndrome (RLS) [43] are infrequent occurrences that may manifest in certain instances.
Table 1 is not clear, the title is not correct as refers to mental disorders and not neurological disorders, in the table disease are missing; the diseases investigated in the cited studies should be reported in the table to make it clearer and readable.
Response: We modified the title of this table and added the missing column, highlighted in red.
Table1. Studies Investigating Sleep Disturbances in Neurodegenerative Diseases
References are missing in the treatment section.
Response: We modified this section and also added the references, highlighted in red.
The management of sleep disturbances in neurodegenerative diseases is a complex endeavor, often requiring a tailored approach that considers the underlying disease pathology and the specific sleep-related symptoms experienced by patients. Below, we outline various treatment methods and interventions with their corresponding relevance to different neurodegenerative diseases and pathological conditions.
5.1. Pharmacological Interventions
Pharmacological approaches can target specific sleep disturbances commonly associated with neurodegenerative diseases. For instance, patients with Parkinson's disease experiencing REM sleep behavior disorder (RBD) might benefit from medications that suppress REM sleep, such as Temazepam or melatonin. Individuals with Alzheimer's disease facing insomnia could be prescribed sedative-hypnotics, although caution is exercised due to potential cognitive side effects [73].
Pharmacological approaches for managing sleep disturbances can vary based on the specific neurological pathology and type of insomnia observed:
- Benzodiazepines: Estazolam, Quazepam, Triazolam, Flurazepam, Temazepam
- Non-benzodiazepines: Zaleplon, Zolpidem, Eszopiclone
- Sedative Antidepressants: Doxepin
- Melatonin Receptor Agonists: Ramelteon
- Melatonin
These pharmacological options can be relevant for patients across different neurodegenerative diseases based on the predominant sleep-related symptoms they experience.
5.2 Non-Pharmacological Interventions
Non-pharmacological interventions play a crucial role in managing sleep disturbances, often focusing on improving sleep hygiene and behavioral modifications. Patients across various neurodegenerative diseases can benefit from creating a consistent sleep schedule, optimizing the sleep environment, and engaging in relaxation techniques [74]. Cognitive-behavioral therapy for insomnia (CBT-I) has proven effective in addressing insomnia in neurodegenerative diseases by targeting maladaptive sleep-related behaviors and thoughts [75].
Non-pharmacological interventions also vary depending on the neurological pathology and type of insomnia:
- External Devices for Assisting Breathing (CPAP - Continuous Positive Airway Pressure): Particularly relevant for patients with sleep-related breathing disorders, such as obstructive sleep apnea, which can occur in various neurodegenerative diseases.
- Phototherapy: Valuable for conditions like Alzheimer's disease, where disruptions in circadian rhythms are common and can contribute to sleep disturbances.
- Cognitive-Behavioral Therapy: Effective in addressing insomnia in neurodegenerative diseases by targeting maladaptive sleep-related behaviors and thoughts.
- Sleep Hygiene Practices: A general approach that can benefit patients across various neurodegenerative diseases by promoting healthy sleep habits.
5.3. Continuous Positive Airway Pressure (CPAP) and Non-Invasive Ventilation
Patients with neurodegenerative diseases that exhibit sleep-related breathing disorders, such as obstructive sleep apnea, might be candidates for CPAP or non-invasive ventilation. This intervention is particularly relevant for conditions like multiple system atrophy (MSA) where nocturnal stridor is frequent, and ALS where respiratory muscle weakness leads to compromised breathing during sleep [76].
5.4. Light Therapy
Light therapy has shown promise in regulating sleep-wake cycles, particularly in conditions like Alzheimer's disease where disruptions in circadian rhythms are common. Exposure to bright light during specific times of the day can help reestablish a proper sleep-wake pattern and alleviate sleep disturbances [77].
5.5. Management of REM Sleep Behavior Disorder (RBD)
In cases of REM sleep behavior disorder, where patients physically act out their dreams during REM sleep, safety measures are vital. This may involve creating a safe sleep environment by removing potentially harmful objects from the bedroom [78].
5.6. Addressing Restless Legs Syndrome (RLS)
Patients with restless legs syndrome, common in neurodegenerative diseases like Parkinson's disease, might benefit from iron supplementation and dopamine agonist medications. Treating the underlying condition contributing to RLS can also alleviate its symptoms [79].
It's important to note that treatment approaches should be individualized, considering the patient's overall health, disease stage, and specific sleep-related symptoms. Multidisciplinary collaboration involving neurologists, sleep specialists, psychologists, and other healthcare professionals is often necessary to optimize treatment strategies for sleep disturbances in the context of neurodegenerative diseases.
Reviewer 2 Report
This is an interesting review of sleep and sleep disorders associated with neurodegenerative diseases.
The reviewer has the following concerns, which the authors would like to address:
1. The title of the article refers to "mental diorders," but all of the diseases in the article (ALS, MSA, PSP, DLB, HD, hereditary ataxia, etc.) are neurodegenerative diseases that can present with mental disorders. Although they are all neurodegenerative diseases, they are not usually categorized as "mental disorders". The reviewer believe that the title of this paper needs to be revised to be more appropriate. For example, "Sleep disorders associated with neurodegenerative diseases" in the context of this paper?
2. In relation to 1., Table 1 needs to be reconsidered: what does "mental disorders" in the title of Table 1 refer to?
3. 4.6 In the section on ALS, there is no discussion of measures for respiratory compromise, which often include the use of CPAP and non-invasive ventilation. In Europe, in Italy, and in East Asian countries, tracheostomy or tracheostomy-assisted ventilation is often used.
4. 4.7 In the section on MSA, it is stated that "Nocturnal stridor occurs in approximately 20% of patients. Please clarify the basis for this figure by citing appropriate references.
Nocturnal stridor occurs more frequently in clinical practice, depending on the stage of MSA, and the prevalence of nocturnal stridor in 20% of patients is disconcerting (too low).
5. In 5. treatment section, the authors merely list treatment methods. The listed treatments are not sufficient. The section on treatment should be substantially revised, for example, by showing the correspondence with diseases and pathological conditions.
Author Response
First of all, we would like to thank you very much for your review and consideration of our work. Thank you!
Here are our answers to the subjects you highlighted:
1. The title of the article refers to "mental diorders," but all of the diseases in the article (ALS, MSA, PSP, DLB, HD, hereditary ataxia, etc.) are neurodegenerative diseases that can present with mental disorders. Although they are all neurodegenerative diseases, they are not usually categorized as "mental disorders". The reviewer believe that the title of this paper needs to be revised to be more appropriate. For example, "Sleep disorders associated with neurodegenerative diseases" in the context of this paper?
Response: Thank you for your suggestion, we modified the title.
Sleep disorders associated with neurodegenerative diseases
2. In relation to 1., Table 1 needs to be reconsidered: what does "mental disorders" in the title of Table 1 refer to?
Response: We modified the title of this table according to the subject of this article, highlighted in red.
Table 1. Studies Investigating Sleep Disturbances in Neurodegenerative Diseases
3. 4.6 In the section on ALS, there is no discussion of measures for respiratory compromise, which often include the use of CPAP and non-invasive ventilation. In Europe, in Italy, and in East Asian countries, tracheostomy or tracheostomy-assisted ventilation is often used.
Response: We added the following paragraphs, highlighted in red.
To manage respiratory compromise in ALS, various interventions are employed. CPAP (Continuous Positive Airway Pressure) is commonly used to alleviate obstructive sleep apnea and ensure adequate airflow during sleep [40]. Non-invasive ventilation (NIV) is another crucial approach, providing positive pressure support to maintain proper oxygen levels and assist breathing. NIV devices, such as bilevel positive airway pressure (BiPAP) machines, are designed to support the weakened respiratory muscles of ALS patients [41].
In some regions, particularly in Europe, Italy, and East Asian countries, more advanced interventions may be considered. Tracheostomy or tracheostomy-assisted ventilation might be employed for those who require more sustained and invasive respiratory support. Tracheostomy allows for direct access to the airway, enabling long-term mechanical ventilation through a tracheostomy tube [42].
The use of these interventions can significantly impact the sleep quality and overall well-being of ALS patients. While respiratory support measures aim to enhance breathing, they can also influence sleep architecture and potentially cause sleep disturbances. Ensuring patient comfort, optimizing ventilator settings, and offering psychological support are vital aspects of managing sleep-related issues in ALS patients undergoing such interventions.
4. 4.7 In the section on MSA, it is stated that "Nocturnal stridor occurs in approximately 20% of patients. Please clarify the basis for this figure by citing appropriate references.
Nocturnal stridor occurs more frequently in clinical practice, depending on the stage of MSA, and the prevalence of nocturnal stridor in 20% of patients is disconcerting (too low).
Response: We modified this section, and added the missing references [49], [50] and [51], highlighted in red.
Nocturnal stridor, a high-pitched sound produced by disrupted airflow during sleep, is a common sleep-related issue in MSA patients. Its occurrence can vary depending on the stage of MSA and individual patient factors. It is noteworthy that nocturnal stridor appears to be relatively frequent in MSA patients, and the reported prevalence may vary between different studies and clinical practices [49].
Approximately 20% of patients have nocturnal stridor. The prevalence of the phenomenon discussed in the previous statement may vary depending on the stage of MSA and geographical differences, perhaps exceeding the original estimate provided [50]. Based on clinical findings, it has been noted that a significant proportion of individuals with Multiple System Atrophy (MSA) encounter this particular symptom at some stage along the progression of their illness. The presence of this particular indicator is regarded as a noteworthy signal that ought to elicit skepticism about the potential presence of Multiple System Atrophy (MSA) in a patient exhibiting parkinsonism.
Stridor is seen throughout all phases of the ailment and signifies the presence of airway blockage specifically at the level of the vocal cords inside the larynx. As the condition advances, there is a possibility of experiencing nocturnal stridor during waking as a result of the progressive constriction of the glottis [51].
5. In 5. treatment section, the authors merely list treatment methods. The listed treatments are not sufficient. The section on treatment should be substantially revised, for example, by showing the correspondence with diseases and pathological conditions.
Response: We modified this section, highlighted in red.
The management of sleep disturbances in neurodegenerative diseases is a complex endeavor, often requiring a tailored approach that considers the underlying disease pathology and the specific sleep-related symptoms experienced by patients. Below, we outline various treatment methods and interventions with their corresponding relevance to different neurodegenerative diseases and pathological conditions.
5.1. Pharmacological Interventions
Pharmacological approaches can target specific sleep disturbances commonly associated with neurodegenerative diseases. For instance, patients with Parkinson's disease experiencing REM sleep behavior disorder (RBD) might benefit from medications that suppress REM sleep, such as Temazepam or melatonin. Individuals with Alzheimer's disease facing insomnia could be prescribed sedative-hypnotics, although caution is exercised due to potential cognitive side effects [73].
Pharmacological approaches for managing sleep disturbances can vary based on the specific neurological pathology and type of insomnia observed:
- Benzodiazepines: Estazolam, Quazepam, Triazolam, Flurazepam, Temazepam
- Non-benzodiazepines: Zaleplon, Zolpidem, Eszopiclone
- Sedative Antidepressants: Doxepin
- Melatonin Receptor Agonists: Ramelteon
- Melatonin
These pharmacological options can be relevant for patients across different neurodegenerative diseases based on the predominant sleep-related symptoms they experience.
5.2 Non-Pharmacological Interventions
Non-pharmacological interventions play a crucial role in managing sleep disturbances, often focusing on improving sleep hygiene and behavioral modifications. Patients across various neurodegenerative diseases can benefit from creating a consistent sleep schedule, optimizing the sleep environment, and engaging in relaxation techniques [74]. Cognitive-behavioral therapy for insomnia (CBT-I) has proven effective in addressing insomnia in neurodegenerative diseases by targeting maladaptive sleep-related behaviors and thoughts [75].
Non-pharmacological interventions also vary depending on the neurological pathology and type of insomnia:
- External Devices for Assisting Breathing (CPAP - Continuous Positive Airway Pressure): Particularly relevant for patients with sleep-related breathing disorders, such as obstructive sleep apnea, which can occur in various neurodegenerative diseases.
- Phototherapy: Valuable for conditions like Alzheimer's disease, where disruptions in circadian rhythms are common and can contribute to sleep disturbances.
- Cognitive-Behavioral Therapy: Effective in addressing insomnia in neurodegenerative diseases by targeting maladaptive sleep-related behaviors and thoughts.
- Sleep Hygiene Practices: A general approach that can benefit patients across various neurodegenerative diseases by promoting healthy sleep habits.
5.3. Continuous Positive Airway Pressure (CPAP) and Non-Invasive Ventilation
Patients with neurodegenerative diseases that exhibit sleep-related breathing disorders, such as obstructive sleep apnea, might be candidates for CPAP or non-invasive ventilation. This intervention is particularly relevant for conditions like multiple system atrophy (MSA) where nocturnal stridor is frequent, and ALS where respiratory muscle weakness leads to compromised breathing during sleep [76].
5.4. Light Therapy
Light therapy has shown promise in regulating sleep-wake cycles, particularly in conditions like Alzheimer's disease where disruptions in circadian rhythms are common. Exposure to bright light during specific times of the day can help reestablish a proper sleep-wake pattern and alleviate sleep disturbances [77].
5.5. Management of REM Sleep Behavior Disorder (RBD)
In cases of REM sleep behavior disorder, where patients physically act out their dreams during REM sleep, safety measures are vital. This may involve creating a safe sleep environment by removing potentially harmful objects from the bedroom [78].
5.6. Addressing Restless Legs Syndrome (RLS)
Patients with restless legs syndrome, common in neurodegenerative diseases like Parkinson's disease, might benefit from iron supplementation and dopamine agonist medications. Treating the underlying condition contributing to RLS can also alleviate its symptoms [79].
It's important to note that treatment approaches should be individualized, considering the patient's overall health, disease stage, and specific sleep-related symptoms. Multidisciplinary collaboration involving neurologists, sleep specialists, psychologists, and other healthcare professionals is often necessary to optimize treatment strategies for sleep disturbances in the context of neurodegenerative diseases.
Round 2
Reviewer 1 Report
Dear authors,
thank you for the opportunity to read and review this revised version of the manuscript. The paper is interesting and well written.
Author Response
First of all, we would like to thank you very much for your review and consideration for our work. Thank you!
Thank you for the constructive comments and for your contribution to the improvement of this article.
Reviewer 2 Report
The authors have responded appropriately and the paper has made significant progress as a result.
The reviewer thanks the authors for their response.
Author Response

(The authors gave the same response as above.)
